# Do Protein Supplementation Levels Influence the Performance of Male Nellore Calves Under a Grazing System at Pre-Weaning?

**DOI:** 10.3390/ani15192913

**Published:** 2025-10-07

**Authors:** Marcos Rocha Manso, Luciana Navajas Rennó, Edenio Detmann, Mário Fonseca Paulino, Sidnei Antônio Lopes, Nicole Stephane de Abreu Lima, Deilen Paff Sotelo Moreno, Román Maza Ortega

**Affiliations:** 1Animal Science Department, Federal University of Viçosa, Viçosa 36570-900, Brazil; marcosrochamanso@gmail.com (M.R.M.); lucianarenno@ufv.br (L.N.R.); detmann@ufv.br (E.D.); mariofonsecapaulino@gmail.com (M.F.P.); sidnei.lopes@ufv.br (S.A.L.); nicole.abreu@ufv.br (N.S.d.A.L.); 2Ruminantia Research Group, Animal Production Department, Faculty of Agricultural and Animal Sciences and Natural Resources, Universidad of the Llanos, Villavicencio 500017, Colombia; 3Agriculture and Livestock Sustainable Research Group, Animal Science Department, Faculty of Agricultural Sciences, University of Pamplona, Pamplona 543050, Colombia

**Keywords:** biopsy, creep-feeding, IGF-1, muscle fibers diameter, nitrogen metabolism

## Abstract

Many studies with beef calves in tropical conditions report increased performance in response to increasing the protein level in the supplement. However, evaluating the use of supplements with an adequate protein proportion and ideal supplementation period for Nellore calves in the tropical grasses can allow establishing supplementation strategies that optimize animal performance and improve bioeconomic efficiency. This study measured the effect of different protein levels in the supplement on male Nellore calves at pre-weaning. Our results showed that decreased protein levels in the supplement, from 300 to 150 g CP/kg DM, do not detrimentally affect performance and efficiency of grazing male Nellore calves.

## 1. Introduction

Managing calves until weaning requires differentiated attention due to its impact on maintenance requirements, calf growth rate (kgs of calf’s weaning), and maintenance costs, which affect biological efficiency and economic utility in beef cattle systems [1,2,3]. Therefore, maximizing growth from the pre-weaning period is fundamental to the success and efficiency of beef cattle production systems.

According to Henriques et al. [4] and Costa e Silva et al. [5], after 90 days since born, milk is not sufficient to meet calf nutritional requirements to support potential growth, which makes it increasingly dependent on pasture. However, tropical grasses cannot be considered a balanced diet because they have several nutrient deficiencies or imbalances that may restrict pasture intake and digestibility, metabolic efficiency, and productive performance of cattle [6].

Therefore, the supplementation of calves during pre-weaning using creep feeding systems, combined with correct pasture management, has been one of the leading nutritional management tools to ensure improved weaning weights [7].

Studies by Poppi and McLennan [8] and Carvalho et al. [9] show that it is only possible to obtain additional gains of 200 to 300 g/d with supplementation in animals fed tropical forages. Thus, studies on creep feeding in the tropics have consistently shown increases in the BW of calves at weaning [10,11,12]

Protein is a critical component of the diet for cattle production systems, and, in growing calves, it is an essential nutrient to maintain skeletal muscle growth. However, tropical pastures have often concentrations below the animal requirements [13]. Additionally, protein is the single most expensive nutrient in the ruminant diet, and its unbalanced inclusion can result in increased production costs, dietary inefficiency, delayed rumen development, and negative environmental impact due to increased excretion of N in urine and feces [14,15].

Among studies evaluating the effects of different protein supplementation levels on the performance of grazing pre-weaning beef calves, some have not showed any changes in performance [14], while others have observed improved performance [10,16]. However, there is a gap in the studies evaluating both the use of supplements with an adequate proportion of protein and the ideal supplementation period for pre-weaning beef calves in tropical pastures. One way to increase the efficiency of protein supplementation during pre-weaning beef calves, both in production and financial terms, would be the strategic and timely provision of protein supplements.

Therefore, we hypothesized that providing a lower level of protein supplementation does not detrimentally affect the productive and nutritional performance of beef calves in a tropical pasture, but rather the efficient use of nitrogen compounds can affect the productivity.

Thus, this study aimed to evaluate the effects of sequential supplementation with different crude protein levels and supplementation period on performance, nutritional and metabolic characteristics, and efficiency of nitrogenous compounds use in grazing male Nellore calves at pre-weaning.

## 2. Materials and Methods

### 2.1. Location and Weather Conditions

The experiment was performed at the Beef Cattle Farm of the Animal Science Department of Federal University of Viçosa, Viçosa-MG, Brazil (20°45′ S, 42°52′ W), from January to June (156 experimental days), which includes both the rainy season and the transition to the dry season. During this period, the average monthly rainfall was 57.8 mm (66.6, 84.0, 81.4, 44.8, 51.4, and 18.6 mm), the mean temperature was 20.7 °C (23.3, 22.7, 21.8, 20.5, 18.2, and 17.7 °C), and the average relative humidity was 80.3% (75.8, 79.1, 79.8, 81.4, 83.5, and 82.2%) for January, February, March, April, May, and June, respectively.

### 2.2. Animals, Experimental Design, Diets and Management

Fifty Nellore suckling male calves non-castrated averaging 3.5 ± 0.1 months and 114 ± 2.4 kg initial body weight (BW), accompanied by their dams (average of 3.5 ± 0.43 years old, 473 ± 8.4 kg BW, and 4.5 ± 0.08 BCS on a scale of one to nine points according to National Research Council (NRC) [17]) were distributed in a completely randomized design in a 2 × 2 factorial arrangement. The number of experimental units was two groups with 12 animals and two groups with 13 animals.

The strategies were as follows: Supplementation period in the initial pre-weaning was 1–78th experimental day and in the final pre-weaning was 79–156th experimental day. Low and high protein levels in the supplement were 150 g CP/kg) and 300 g CP/kg), respectively, resulting in four feed strategies: LO–LO, LO–HI, HI–LO, and HI–HI. The randomization accounted for dams’ parity order and milk yield.

The cow-calf pairs were allocated into eight pastures of *Urochloa decumbens* (5.0 ha and 6.25 cow-calf/pasture), previously deferred and equipped with feeders and dispenser water, allowing free access to water and feed. All pastures presented similar forage quality, shade, and water quality and availability. The cows-calf grazed one paddocks using the continuous stocking method.

The supplements (as-fed) were composed of corn meal, soybean meal, wheat meal, molasses, and mineral mix (Table 1), and provided daily at 1100 h in a creep feeding system at 6 g/kg of BW. This amount corresponds to approximately 32% and 64% of the dietary requirements of CP for LO and HI, respectively, for Zebu male calf under grazing conditions with BW of 200 kg and an expected gain of 0.8 kg/day [18]. Calves were weighed at 0600 h every 26 days without fasting to adjust the amount of supplement to be provided.

### 2.3. Forage Samples and Nutritional Performance

Aiming to obtain a representative sample at the beginning, middle and end of each experimental subperiod, the forage chemical composition was assessed by hand-plucked samples collected every 13 days. This was based on the identification of the places of intake and the parts of the plant selected by the animals, simulating the calves’ grazing as closely as possible [19]. A second forage sample was collected every 26 days from each pasture to estimate total forage dry matter (DM) and potentially digestible DM (pdDM). Forage was clipped about 1 cm above the soil surface in four randomly chosen locations using a 0.5 × 0.5 m metal frame. All samples were partially dried in a forced-air oven at 60 °C for 72 h, then ground in a Wiley knife mill to pass through a 2-mm screen. Subsequently, half of each ground sample was reground to pass through a 1-mm screen and stored in plastic pots until analysis.

A 10-day intake-digestibility trial was carried out in each pre-weaning period, starting on the 39th and 118th day of the experiment. From the first to the eighth day of each trial, chromium oxide (Cr_2_O_3_) was used as an external marker to estimate fecal excretion with a dosage of 10 g per animal. The initial five days were used for the animal’s adaptation to the Cr_2_O_3_ (stabilization of marker excretion). The indigestible neutral detergent fiber (iNDF) served as the internal marker to estimate forage DM intake. Fecal samples (approximately 200 g) were collected immediately after defecation on pasture when animals were herded or taken directly from the rectum of animals into a handling facility (at amounts of approximately 200 g). This was carried out on the last four days of the trial, at varying times: day 6 at 1800 h, day 7 at 1400 h, day 8 at 1000 h, and day 9 at 0600 h. Each sample was labeled, partially dried in an oven at 55 °C for 72 h, and ground as previously described.

On day ten of the digestibility trial, spot urinary samples were collected using a plastic container during spontaneous urination, taken 4 h before (0700 h) and 4 h after (1500 h) supplying the supplement, according to Da Silva Júnior’s et al. [20] recommendations. After collection, 10 mL of urine were diluted in 40 mL of H_2_SO_4_ (0.036 N) and stored at −20 °C for later laboratory analysis of creatinine, urea, uric acid, and allantoin concentrations.

Dams were milked on days 50 and 130 of the experiment to estimate the quantity and composition of daily milk intake by calves, using a controlled suckling protocol. Calves were separated from their dams at 1500 h to ensure milk depletion, with cows returning to pasture and calves remaining in the shed. At 1730 h, calves were reunited with their dams and allowed to suckle for 30 min. At 1800 h, calves were once again separated until the following morning. At 0600 h on the next day, cows were mechanically milked immediately after an injection of 2 mL of oxytocin (10 IU/mL; Ocitovet^®^, Vet&Cia Animal Health, São Paulo, Brazil) into the mammary artery, and the milk yield was weighed immediately after milking. The exact time when the milking of each cow ended was recorded, and the milk yield was standardized to a 24 h yield.

Individual samples of 50 mL of milk were taken for analyses of protein, fat, lactose, and total solids. Samples were stored at 4 °C in a refrigerator using a bronopol tablet per sample as a preservative. Milk samples were analyzed using spectroscopy (Foss MilkoScan FT120, Hillerød, Denmark). Milk production was corrected to 4% of fat (Milk_4%_) according to the NRC [17]:Milk_4%_ (kg) = (0.4 × MP) + [15 × (FP × MP/100)]
where MP = milk production (kg/day); and FP = fat production (kg/day).

### 2.4. Blood Samples and Measurements

On days 75 and 150 of the experiment period, blood samples were collected from all calves without fasting to quantify the concentrations of urea, glucose, triglycerides, albumin, total proteins, and insulin-like growth factor type 1 (IGF-1). Samples were collected at 0700 h via jugular venipuncture before supplement feeding using vacuum tubes with sodium fluoride and EDTA as a glycolytic inhibitor and anticoagulant (BD Vacutainer^®^ Fluoride/EDTA, São Paulo, Brazil) for glucose analysis and vacuum tubes with clot and separator gel (BD Vacuntainer^®^, SST II Advance, São Paulo, Brazil) for the other analyses. The blood samples were centrifuged at 2500× *g* for 15 min, and the serum and plasma were subsequently stored at −20 °C for further analysis.

### 2.5. Skeletal Muscle Collection and Histological Analyses

On days 75 and 150 of the experiment, biopsies of skeletal muscle tissue were performed from all calves to obtain samples of the Longissimus dorsi muscle between the 12th and 13th ribs to assessment the number and diameter of muscle fibers. Briefly, this procedure was carried out with the appropriate veterinary care and, the area was cleaned with 70% ethanol, and then the incision was made 10 min after local anesthesia (Lidocaine 2%, LidoVet, Bravet, Rio de Janeiro, Brazil). After sample collection, the site was sutured, and animals received antibiotics and anti-inflammatory treatment, and sutures removed two weeks later. A 2 cm^3^ skeletal muscle samples were washed with sterile saline solution (0.9% NaCl) subsequently fixed in a 4% paraformaldehyde solution at room temperature for 24 h. After incubation, the samples were dehydrated in ethanol solutions of increasing concentration (70, 80, 90, and 100%) for 2 h, diaphorized in xylene for 1 h, and embedded in paraffin. Tissue sections were made in 4 µm sections and stained with toluidine blue. A total of 30 images (magnification = 10×, scale bar = 100 μm, image resolution = 1600 × 1200 pixels) were obtained from each animal using an Olympus BX50 microscope with attached camera CMOS 1.3 MP BioCAM (Takachiho, Miyazaki, Japan). The number and diameter of muscle fibers were analyzed using ImageJ 1.54 software [21].

### 2.6. Productive Performance and Carcass Characteristics

At the beginning and end of each supplementation period, calves were weighed after fasting for 14 h to quantify the average daily gain (ADG) and final body weight (FBW). Simultaneously, the subcutaneous fat thickness over the Longissimus muscle (SFTL) between the 12th and 13th ribs, and subcutaneous fat thickness over the rump (SFTR) between the ischium and pubis, were measured by ultrasound (Aloka SSD 500; 3.5-MHz linear probe; Aloka Co, Tokyo, Japan). Vegetable oil was used to ensure adequate acoustic contact. Images were analyzed using the BioSoft Toolbox^®^ II 4.0 for Beef software (Biotronics Inc., Ames, IA, USA).

### 2.7. Analytical Procedures

Samples of forage, feces, and supplements (ground to 1 mm) were analyzed for dry matter (DM—dried overnight at 105 °C; method INCT-CA G-03/1), mineral matter (MM—complete combustion in a muffle furnace at 600 °C for 4 h; method INCT-CA M-001/1), crude protein (CP—Kjeldahl procedure; method INCT-CA N-001/1), ether extract (EE—Randall procedure; method INCT-CA G-005/1), neutral detergent insoluble fiber (NDF; method INCT-CA F-002/1) corrected for ash (NDIA—neutral detergent insoluble ash; method INCT-CA M-002/1) and protein (NDIP—neutral detergent insoluble protein; method INCT-CA N-004/1), residue (apNDF; using a heat-stable α-amylase, omitting sodium sulfite and correcting for residual ash and protein), and feces samples were also analyzed for chromium concentration (nitroperchloric digestion and atomic absorption spectrophotometry; INCT method -CA M-005/1), according to the procedures proposed by Brazilian National Institute of Science and Technology in Animal Science (INCT-CA) [22]. The content of iNDF in samples of feces, forage, and supplement (ground to 2 mm) was estimated as the residual NDF remaining after 288 h of ruminal in situ incubation using F57 filter bags (Ankom Technology Corp., Macedon, NY, USA).

The quantification of non-fibrous carbohydrates (NFC) was performed according to Hall [23] as follows:NFC = 100 − (%CP + %apNDF + %EE + %MM)

The fecal DM excretion was estimated using the chromic oxide marker, based on the ratio between the amount of chromium administered and its concentration in the feces. Individual DM intake (DMI) was estimated using iNDF as an internal marker and calculated by the following equation:DMI = [(FE × iNDFF − iNDFS)/iNDFP] + ISI + MI
where FE = fecal DM excretion (kg/day); iNDFF = concentration of iNDF in the feces (kg/kg); iNDFS = concentration of iNDF in the supplement (kg/kg); iNDFP = amount of iNDF form pasture (kg/kg); ISI = individual supplement intake (kg/day); and MI = milk intake (kg/day).

The pdDM in forage mass on pasture was estimated using the following equation:pdDM = 0.98 × (100 − NDF) + (NDF − iNDF)
where 0.98 is the true digestibility coefficient of intracellular content; NDF is forage content of neutral detergent fiber; and iNDF is forage content of indigestible neutral detergent fiber.

Urinary concentrations of creatinine (K067, Bioclin^®^ Quibasa, Belo Horizonte, Brazil), urea (K056, Bioclin^®^ Quibasa, Belo Horizonte, Brazil), and uric acid (K139, Bioclin^®^ Quibasa, Belo Horizonte, Brazil) were measured by a colorimetric kinetic, colorimetric enzymatic, and fixed-time kinetic methods, respectively, using the BS-200E automatic biochemistry analyzer (Mindray BS200E, Shenzhen, China). The daily urinary volume was calculated using the ratio between the daily creatinine excretion (CE) and creatinine concentration in urine, according to the following equation proposed by Costa e Silva et al. [24]:CE (g/day) = 0.0345 × SBW^0.9491^
where SBW = shrunk body weight.

The total excretion of purine derivatives was calculated by sum of the allantoin and uric acid excreted in the urine, obtained by the product of their concentration in the urine and the daily urinary volume. Absorbed purines (PA, mmol/day) were calculated from the excretion of purine derivatives (PD, mmol/day) through the following equation:AP = (PD − 0.301 × BW^0.75^)/0.80
where 0.80 is the recovery of purines absorbed as purine derivatives and 0.301 × BW^0.75^ is the endogenous contribution to purine excretion [25].

Ruminal synthesis of nitrogenous compounds (Nmic, g Nmic/day) was calculated as a function of absorbed purines (AP, mmol/day) using the following equation described by Barbosa et al. [25]:Nmic: 70 × AP/(0.93 × 0.137 × 1000)
where 70 is purine N content (mg N/mol); 0.93: true digestibility of bacterial purines, and 0.137: N purine: total N ratio in bacteria.

The microbial efficiency was obtained by the ratio between production of microbial crude protein (CPmic), expressed in grams, and the consumed amount of digested organic matter (DOM) expressed in kilograms.

Blood urea (K056, Bioclin^®^ Quibasa, Belo Horizonte, Brazil), glucose (K082, Bioclin^®^ Quibasa, Belo Horizonte, Brazil), and triglycerides (K117, Bioclin^®^ Quibasa, Belo Horizonte, Brazil) concentrations were quantified by enzymatic-colorimetric methods. Albumin (K040, Bioclin^®^ Quibasa, Belo Horizonte, Brazil) and total proteins (K031, Bioclin^®^ Quibasa, Belo Horizonte, Brazil) using colorimetric methods in an automatic biochemistry analyzer (Mindray BS200E, Shenzhen, China). Globulins concentrations were calculated as the difference between the analyzed total protein and albumin content. Serum urea nitrogen (SUN) was estimated as 46.67% of the total serum urea. IGF-1 (313231, Liaison^®^) in Liaison^®^ analyzer (DiaSorin^®^, Saluggia, Italy) was analyzed by the chemiluminescence method.

### 2.8. Statistical Analysis

The experiment was conducted and analyzed in a completely randomized design with a duple error structure. The MIXED procedure of the SAS 9.4 software was used for all statistical analyses. To determine the effect of the variables measured during the initial pre-weaning, two treatments were evaluated as a simple comparison between calves that received low and high protein levels in the supplement. To determine the effect of the variables measured during the final pre-weaning, the sums of squares of the four treatments were decomposed using orthogonal contrasts to test the interaction and independent effects of the protein level in the supplement in a 2 × 2 factorial arrangement (two protein levels in the supplement in initial pre-weaning; 150 g CP/kg [LO] and 300 g CP/kg [HI] treatments, respectively; and two in final pre-weaning (LO and HI treatments). The effect of treatment on all variables measured was evaluated by ANOVA, according to the following mathematical model:Yijk = μ + αi +βj +(αβ)ij + e(i)j +ε(ij)k
where Yijk = observations of individual k on paddock j under treatment i; μ = overall mean; αi = fixed effect of supplementation protein levels; βj = fixed effect of initial pre-weaning and final pre-weaning; (αβ)ij = interaction effect between supplementation protein levels and supplementation period at pre-weaning; e(i)j = random error, unobservable, associate to each j paddock under treatment i, assumed to be normally and independently distributed (NID; 0, σe^2^); and ε(ij)k: random error, unobservable, associate to each k observation on j paddock under treatment i, assumed to be NID (0, σe^2^).

The choice of the most appropriate covariance structure was based on the lowest value of the corrected Akaike information criterion. The degrees of freedom were estimated according to the Kenward–Roger method. The data showed normality by the Shapiro–Wilk test and homoscedasticity through the Bartlett test. Differences were considered significant at *p*-value ≤ 0.05.

## 3. Results

There was no interaction between protein levels and pre-weaning period for any of the variables evaluated in this study; therefore, only the main effects are discussed separately.

### 3.1. Intake, Digestibility, and Nitrogen Balance

The overall means of DM and pdDM mass were 4.59 and 2.96 t/ha in initial pre-weaning and 3.74 and 2.29 t/ha in final pre-weaning. These represent a potential use of 64.5 and 61.4% of the forage mass for initial and final pre-weaning, respectively. The average CP content of the pasture consumed by the animals during the experimental period was 81 ± 1.7 g CP/kg DM. The CP content of the supplements was 167 ± 8.52 and 324 ± 10.3 g CP kg/DM for LO and HI, respectively (Table 1).

The average milk yield, milk_4%_, and composition were not affected by protein levels in the supplement consumed by male calves (Table 2). In initial pre-weaning, protein levels in the supplement did not affect the intake of DM, forage DM (FDM), milk DM (MDM), organic matter (OM), EE, apNDF, iNDF, and DOM (Table 3). However, higher CP intakes and CP:DOM ratios (*p*-value < 0.05) were evident in the HI calves compared with the LO calves. In contrast, a higher NFC intake (*p*-value < 0.05) was observed in LO than in HI calves.

In final pre-weaning, no effect of protein levels was observed on the intake of DM, FDM, MDM, OM, EE, apNDF, iNDF, Digested NDF (DNDF), and DOM (Table 3). Nevertheless, there was a higher intake (*p*-value < 0.05) of CP (0.74 ± 0.025 kg/day) and CP: DOM (367.5 ± 7.35 g CP/kg DOM) in HI calves compared to LO calves (0.54 ± 0.025 CP kg/day and 267.5 g ± 7.35 CP/kg DOM). A higher intake of NFC (*p*-value < 0.05) was observed for LO calves (mean = 1.17 ± 0.045 kg/day) in comparison with the HI calves (1.03 ± 0.045 kg/day).

The protein levels in the supplement did not affect the variables related to the total digestibility of the DM (0.694 ± 0.0182 and 0.514 ± 0.0251 g/g), OM (0.743 ± 0.0156 and 0.565 ± 0.0227 g/g), CP (0.740 ± 0.0240 and 0.664 ± 0.0245 g/g), EE (0.874 ± 0.0128 and 0.711 ± 0.0406 g/g), apNDF (0.613 ± 0.0275 and 0.411 ± 0.0140 g/g), and NFC (0.8 ± 0.0126 and 0.647 ± 0.0456 g/g) in initial pre-weaning and final pre-weaning respectively (Table 4).

In initial pre-weaning, the protein levels in the supplement did not affect the synthesis of nitrogenous compounds in the rumen (NMIC), microbial nitrogen to consumed nitrogen (MICNR) ratio, efficiency of microbial synthesis (EMS), fecal nitrogen excretion (FNE), nitrogen balance (NB), or efficiency of nitrogen utilization (EFNU) (Table 5). Nonetheless, the HI calves had higher (*p*-value < 0.05) nitrogen intake (NI), concentrations of SUN, urinary urea nitrogen (UUN), and urinary excretion of urea nitrogen (UEUN) than LO calves.

In final pre-weaning, no differences between treatments were observed for NMIC, EMS, FNE, NB, or EFNU (Table 4). However, the MICNR was lower (*p*-value < 0.05) in the HI calves (average = 0.26 ± 0.048 g/g) than in the LO calves (0.42 ± 0.048 g/g). As observed in initial pre-weaning, supplemented animals with higher protein levels in final pre-weaning also had higher (*p*-value < 0.05) NI (mean = 119 g/day in HI calves and 86 g/day in LO calves), SUN concentration (HI: 16.8; LO: 10.9 mg/dL), and UUN (HI: 48.1; LO: 21.9 g/day), and UEUN (HI: 59.9; LO: 34.9 g/day) (Table 5).

### 3.2. Hormone and Metabolite Concentrations

In initial pre-weaning, the protein levels in the supplement did not affect the blood concentrations of IGF-1, glucose, triglycerides, total protein, or globulins in the animals (Table 6). Only albumin concentration was higher (*p*-value < 0.05) for the HI calves (3.42 ± 0.115 g/dL) (Table 6). In final pre-weaning, there was no difference in the blood concentrations of metabolites and hormones evaluated between treatments.

### 3.3. Productive Performance and Carcass Characteristics

In initial pre-weaning, the protein levels in the supplement did not affect the FBW, ADG, SFTL, or SFTR in the calves (Table 7). The mean of both treatments for ADG and FBW were 0.99 ± 0.023 kg/day and 192.0 ± 1.41 kg of BW between the treatments. Likewise, there was no effect of the protein levels in the supplement on the number and diameter of muscle fibers (Figure 1A, B, and C, respectively).

In final pre-weaning, there was no effect of protein levels in the supplement on FBW, ADG, SFTL, or SFTR (Table 7). The male calves presented on average, 0.89 ± 0.032 kg/day of ADG and 252.0 ± 5.29 kg of BW across the treatments. The muscle fiber number and diameter were not affected by protein levels in the supplement (Figure 2A–C).

## 4. Discussion

The forage consumed by male calves presented an average CP content of close to the value of 70–80 g of dietary CP/kg DM. This value is consistent with the suggestion by Lazzarini et al. [26] for the minimum limit for the use of the fibrous carbohydrates from forage. Some studies in cattle that are fed tropical forage have suggested that protein supplementation can increase the dietary CP content to nearly 100 g CP/kg DM, optimizing forage intake [27]. Thus, at this level of dietary CP, the nitrogen compound requirements of the rumen microorganisms are met, and further benefits of forage degradation would not be observed.

In this study, the average dietary CP content was 140.2 g of CP/kg DM for LO-LO and HI-LO calves and 187.1 g of CP/kg for LO-HI and HI-HI calves. These values differ from the CP content required by the ruminal microbiota (145 g CP/kg DM), to maximize the consumption of fibrous carbohydrates [28], which may justify the similar intake of forage and apNDF digestibility between treatments.

The animals that received higher levels of protein in the supplement across both supplementation periods at pre-weaning had a higher intake of CP. However, MDM, FDM, and supplement DM intake did not vary between treatments. This lack of difference may be attributed to possible energy–protein ratio imbalance. In addition, the higher intake of NFC in the animals receiving a lower level of protein in the supplement in both supplementation periods at pre-weaning may be due to the lower CP content in the supplement, as only the CP intake showed a difference between them.

The similar CP digestibility between treatments in initial pre-weaning can be attributed to the higher intake of easily digestible components, as the milk intake by calves averaged 7.10 ± 0.331 and 5.43 ± 0.444 kg/day for initial and final pre-weaning, respectively. In addition, the best nutritive value and mass of forage in initial pre-weaning allowed animals to be more selective to satisfy their potential nutrient utilization rates. In contrast, in the final pre-weaning, the higher CP digestibility in the HI treatment can be explained by the higher intake of CP via supplement and lower proportion of milk in the animals’ diet, which promotes a lower proportion of the fecal metabolic fraction of nitrogenous compounds [29].

The higher levels of SUN in animals from HI treatments are due to the increase in dietary NI associated with higher rates of ammonia transfer from the rumen into the blood. Thus, higher levels of SUN caused the greatest elimination of nitrogen compounds via urine (UUN and UEUN) in these animals, which can lead to an increased the risk of nitrate (NO_3^−^_) leaching and ammonia (NH_3_) and nitrous oxide (N_2_O) emissions [30]; this may indicate excess protein in the diet or a fraction of N, whose efficiency of use can be optimized and lead to an increase in the EMS. Therefore, better synchrony of the ratio of protein to energy would be necessary.

Decreasing the CP level in the supplement did not have a detrimental effect on the nitrogen balance and EFU in the animals, which may suggest that the PB flow of 150 g via supplement was sufficient to allow an adequate balance of degradable protein in the rumen, favoring the contribution of microbial protein. This finding could be attributed to all treatments presenting a CP to DOM ratio greater than the 192 g PB/MOD suggested by Valadares Filho et al. [18] for an animal weighing 250 kg BW and expected gain of 900 g/day, exhibiting inadequate protein to energy ratio and affecting the body’s nitrogen deposition and utilization efficiency. Thus, the treatments promoted similar anabolic effects, a conclusion supported by the similar blood IGF-1 concentrations and the productive performance of the animals. IGF-1 is a hormone produced by adipose, muscle, and liver tissues and an endocrine regulator of muscle growth in cattle, which, in addition to its independent action, is a critical link between growth hormone (GH) and the metabolic process of growth [31,32], particularly in skeletal muscle [33].

The highest concentration of albumin in the HI calves in initial pre-weaning, may be explained by the greater CP dietary intake by these animals. According to Lawrence et al. [34], the albumin concentrations in blood may be associated with amino acid and nutrient availability. On the other hand, our results suggest that the diets provided the animals with similar nutrient intake, which may explain the absence of differences in the other metabolites evaluated from blood. It is important to highlight that all animals received a supplementation level of 6 g/kg BW.

Similar responses in ADG, FBW, SFTL and SFTR between treatments can be attributed to the lack of effect of protein levels in the supplement on DM, DMF, DMM, OM, and DOM intakes and nutrient digestibility, except for CP in final pre-weaning. Further-more, the absence of effect of protein levels can also be justified by the reduction of urinary urea nitrogen excretion and an increase in the gastrointestinal urea entry rate, in parallel with an upregulation of the ruminal urea transporter B. This indicates a higher urea transport capacity to the rumen from animals receiving lower protein levels in the supplement, resulting in a similar body N retention between treatments [35].

Regarding the number and diameter of muscle fibers, our data revealed that protein levels in the supplement during the pre-weaning period did not increase the number and diameter of myofibers in skeletal muscle. According to Bonnet et al. [36], Du et al. [37], and Duarte et al. [38], the total number of muscle fibers is set by the end of the second trimester of gestation. Therefore, greater nutrient availability due to maternal supplementation at mid-gestation may increase the number of myofibers in the skeletal muscle of calves. On the other hand, Du et al. [39] suggested that the last trimester of gestation is when fetal skeletal muscle mass increases, mainly due to muscle fiber hypertrophy. Thus, our results indicate that reducing protein levels in the supplement in the pre-weaning period does not contribute to reduced myogenesis in male calves’ skeletal muscle, which may explain the similarity in the number and diameter of myofibers in skeletal muscle across treatments.

## 5. Conclusions

The decrease in protein levels in the supplement from 300 to 150 g CP/kg DM in grazing male Nellore calves under creep feeding at 6 g/kg BW at pre-weaning does not detrimentally affect performance, nutritional, or metabolic characteristics, or the efficiency of using nitrogenous compounds. Therefore, supplementation with 150 g CP/kg DM in the amount of 6 g/kg BW for male beef calves at pre-weaning is recommended.

## Figures and Tables

**Figure 1 animals-15-02913-f001:**
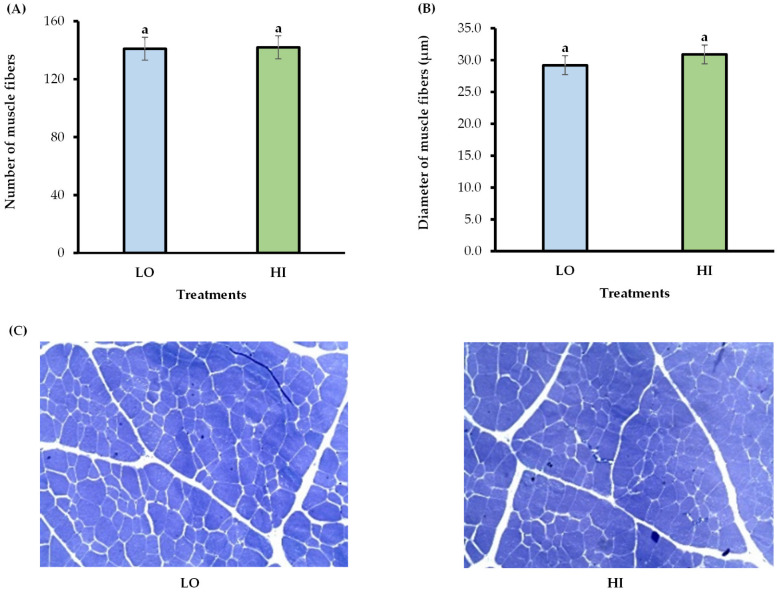
Effect of sequential supplementation with different crude protein levels on number and diameter muscle fibers (**A**,**B**) and Longissimus dorsi muscle microscopy images (**C**) in grazing male Nellore calves in the initial pre-weaning period. LO = animals received 6 g/kg of BW of supplement with 150 g CP/kg DM; HI = animals received 6 g/kg of BW of supplement with 300 g CP/kg DM. The samples were collected per biopsies of longissimus dorsi muscle. The bars represent means ± SEM (*n* = 25). Means over the bars are followed by letters (*p*-value < 0.05). The cell number is per reading area (10×).

**Figure 2 animals-15-02913-f002:**
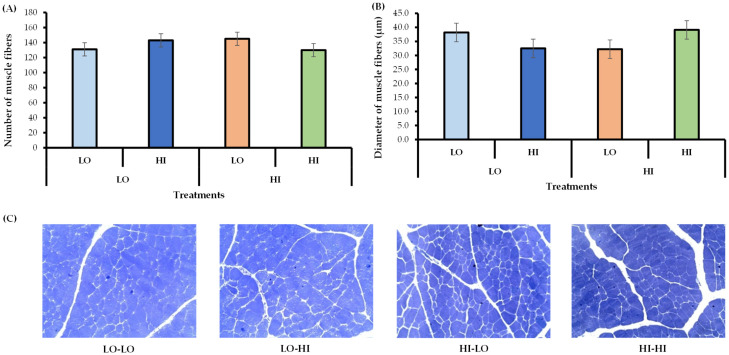
Effect of sequential supplementation with different crude protein levels on number and diameter muscle fibers (**A**,**B**) and Longissimus dorsi muscle microscopy images (**C**) in grazing male Nellore calves in the final pre-weaning period. LO-LO = animals received 6 g/kg of BW of supplement with 150 g CP/kg DM in the initial (1–78th experimental day) and final (79–156th experimental day) pre-weaning; LO-HI = animals received 6 g/kg of BW of supplement with 150 g CP/kg DM in the initial pre-weaning and supplement with 300 g CP/kg DM in the final pre-weaning; HI-LO = animals received 6 g/kg of BW of supplement with 300 g CP/kg DM in the initial pre-weaning and supplement with 150 g CP/kg DM in the final pre-weaning; HI-HI = animals received 6 g/kg of BW of supplement with 300 g CP/kg DM in the initial and final pre-weaning. The samples were collected per biopsies of longissimus dorsi muscle. The bars represent means ± SEM (*n* = 12). The cell number is per reading area (10×).

**Table 1 animals-15-02913-t001:** Ingredients and chemical composition of supplement and forage consumed by the animals during the experimental period.

Item	Low Protein Supplement	High Protein Supplement	Forage ^4^	Forage ^5^	Forage ^6^
Ingredient % (as fed basis)					
Corn meal	18.0	55.0	–	–	–
Wheat bran	55.0	18.0	–	–	–
Soybean meal	19.0	19.0	–	–	–
Molasses	3.0	3.0	–	–	–
Mineral mix ^1^	5.0	5.0	–	–	–
Chemical composition (g/kg DM)					
Dry matter	883.0	881.0	274.0 ± 2.90	376.0 ± 13.80	314.0 ± 15.2
Organic matter	926.0	920.0	924.0 ± 3.30	929.0 ± 2.30	929.0 ± 3.00
Crude protein	167.0	324.0	88.4 ± 2.10	74.4 ± 1.20	76.0 ± 2.70
Ether extract	26.5	24.1	15.2 ± 0.90	10.5 ± 0.60	15.8 ± 0.90
Non fibrous carbohydrates	524.0	403.0	188.0 ± 5.20	133.0 ± 7.90	154.0 ± 10.40
apNDF ^2^	202.0	175.0	632.0 ± 4.50	711.0 ± 6.30	684.0 ± 13.00
iFDN ^3^	38.6	36.6	206.0 ± 10.60	330.0 ± 4.2	254.0 ± 16.30

^1^ Mineral mix = 50% dicalcium phosphate, 47.2% sodium chloride, 1.5% zinc sulfate, 0.7% copper sulfate, 0.05% cobalt sulfate, 0.05% potassium iodate, and 0.5% manganese sulfate. ^2^ apNDF, neutral detergent fiber corrected for ash and protein residue. ^3^ iNDF, indigestible neutral detergent fiber. ^4, 5^ Forage = Means ± standard error of the mean of sample obtained by hand-plucking in the digestibility assay in each period. ^6^ Forage = Means ± standard error of the mean obtained by hand-plucking during each experimental period.

**Table 2 animals-15-02913-t002:** Milk yield and composition of cows according to the protein level delivered to the male calves.

Item	Initial Pre-Weaning ^2^				SEM ^3^	IP	*p*-Value ^4^		
	LO		HI				Overall		
	Final Pre-Weaning						IP	FP	IP × FP
	LO	HI	LO	HI					
	kg/day								
Yield milk	7.12		7.07		0.331	0.930	–	–	–
Milk_4%_ ^1^	8.16		8.12		0.450	0.942	–	–	–
Fat	0.36		0.35		0.023	0.951	–	–	–
Protein	0.23		0.24		0.019	0.851	–	–	–
Lactose	0.33		0.33		0.024	0.964	–	–	–
Total solids	0.99		0.99		0.053	0.915	–	–	–
Yield milk	5.41	5.08	5.56	5.66	0.444		0.472	0.820	0.663
Milk_4%_	6.85	6.13	6.79	7.24	0.661		0.493	0.857	0.451
Fat	0.31	0.27	0.30	0.33	0.041		0.529	0.878	0.419
Protein	0.19	0.19	0.21	0.22	0.020		0.274	0.881	0.604
Lactose	0.23	0.22	0.25	0.24	0.026		0.475	0.765	0.811
Total solids	0.77	0.73	0.81	0.84	0.075		0.370	0.957	0.649

^1^ Milk_4%_: milk production corrected to 4% of fat. ^2^ LO = animals received 6 g/kg of BW of supplement with 150 g CP/kg DM; HI = animals received 6 g/kg of BW of supplement with 300 g CP/kg DM. ^3^ SEM = standard error of means. ^4^ IP = effect protein level in the supplement at initial pre-weaning; FP = effect protein level in the supplement at final pre-weaning; and IP × FP = interaction effect between supplementation levels at initial pre-weaning × final pre-weaning.

**Table 3 animals-15-02913-t003:** Voluntary intake of male Nellore calves under grazing system receiving different protein levels in the supplement at pre-weaning.

Item	Initial Pre-Weaning ^7^				SEM ^8^	IP	*p*-Value ^9^		
	LO		HI				Overall		
	Final Pre-Weaning						IP	FP	IP × FP
	LO	HI	LO	HI					
	kg/day								
Total DM ^1^	3.05		2.99		0.148	0.744	–	–	–
Forage DM	1.27		1.38		0.136	0.554	–	–	–
Milk DM	0.99		0.99		0.055	0.938	–	–	–
OM ^2^	2.89		2.82		0.137	0.718	–	–	–
Crude protein	0.48		0.56		0.025	0.011	–	–	–
Ether extract	0.4		0.39		0.027	0.865	–	–	–
apNDF ^3^	0.96		0.98		0.082	0.851	–	–	–
NFC ^4^	1.06		0.89		0.058	0.044	–	–	–
Indigestible NDF ^5^	0.29		0.31		0.035	0.697	–	–	–
Digested NDF	0.58		0.61		0.062	0.687	–	–	–
Digested OM	2.12		2.09		0.098	0.813	–	–	–
CP: DOM (g/kg) ^6^	227.0		269.0		7.05	0.006	–	–	–
Total DM	3.71	3.87	3.92	4.04	0.214		0.461	0.583	0.927
Forage DM	1.72	1.91	1.88	1.96	0.202		0.651	0.570	0.809
Milk DM	0.78	0.74	0.82	0.85	0.076		0.370	0.956	0.650
Organic matter	3.48	3.64	3.69	3.80	0.207		0.436	0.562	0.932
Crude protein	0.52	0.73	0.55	0.75	0.025		0.353	0.001	0.916
Ether extract	0.36	0.33	0.36	0.38	0.049		0.524	0.851	0.486
apNDF	1.48	1.57	1.57	1.64	0.189		0.680	0.678	0.966
NFC	1.12	1.02	1.21	1.03	0.045		0.292	0.027	0.438
Indigestible NDF	0.62	0.70	0.66	0.68	0.068		0.844	0.499	0.667
Digested NDF	0.64	0.61	0.62	0.68	0.102		0.816	0.908	0.669
Digested OM	2.04	1.96	2.02	2.11	0.103		0.549	0.988	0.455
CP: DOM (g/kg)	260.0	375.0	275.0	360.0	7.35		0.976	<0.001	0.130

^1^ DM = dry matter. ^2^ OM = organic matter. ^3^ apNDF = neutral detergent fiber corrected for ash and protein residue. ^4^ NFC = non fibrous carbohydrates. ^5^ NDF = neutral detergent fiber. ^6^ CP: DOM = crude protein and digested organic matter ratio. ^7^ LO = animals received 6 g/kg of BW of supplement with 150 g CP/kg DM; HI = animals received 6 g/kg of BW of supplement with 300 g CP/kg DM. ^8^ SEM = standard error of means. ^9^ IP = effect protein level in the supplement at initial pre-weaning; FP = effect protein level in the supplement at final pre-weaning; and IP × FP = interaction effect between supplementation levels at initial pre-weaning × final pre-weaning.

**Table 4 animals-15-02913-t004:** Total digestibility of male Nellore calves under grazing system receiving different protein levels in the supplement at pre-weaning.

Item	Initial Pre-Weaning ^3^				SEM ^4^	IP	*p*-Value ^5^		
	LO		HI				Overall		
	Final Pre-Weaning						IP	FP	IP × FP
	LO	HI	LO	HI					
	g/g								
Dry matter	0.695		0.692		0.0182	0.912	–	–	–
Organic matter	0.740		0.745		0.0156	0.823	–	–	–
Crude protein	0.724		0.756		0.0240	0.391	–	–	–
Ether extract	0.877		0.871		0.0128	0.733	–	–	–
apNDF ^1^	0.603		0.622		0.0275	0.634	–	–	–
NFC ^2^	0.800		0.800		0.0126	0.803	–	–	–
Dry matter	0.545	0.496	0.503	0.510	0.0251		0.613	0.474	0.342
Organic matter	0.592	0.547	0.554	0.568	0.0227		0.757	0.551	0.287
Crude protein	0.647	0.702	0.606	0.700	0.0245		0.454	0.045	0.496
Ether extract	0.747	0.666	0.726	0.706	0.0406		0.817	0.298	0.515
apNDF	0.434	0.394	0.400	0.415	0.0140		0.696	0.456	0.141
NFC	0.699	0.611	0.659	0.619	0.0456		0.746	0.242	0.642

^1^ apNDF = neutral detergent fiber corrected for ash and protein residue. ^2^ NFC = non fibrous carbohydrates. ^3^ LO = animals received 6 g/kg of BW of supplement with 150 g CP/kg DM; HI = animals received 6 g/kg of BW of supplement with 300 g CP/kg DM. ^4^ SEM = standard error of means. ^5^ IP = effect protein level in the supplement at initial pre-weaning; FP = effect protein level in the supplement at final pre-weaning; and IP × FP = interaction effect between supplementation levels at initial pre-weaning × final pre-weaning.

**Table 5 animals-15-02913-t005:** Synthesis and efficiency of use of nitrogenous compounds of male Nellore calves under grazing system receiving different protein levels in the supplement at pre-weaning.

Item ^1^	Initial Pre-Weaning ^2^				SEM ^3^	IP	*p*-Value ^4^		
	LO		HI				Overall		
	Final Pre-Weaning						IP	FP	IP × FP
	LO	HI	LO	HI					
NI (g/day)	76.10		89.10		2.517	0.011	–	–	–
NMIC (g/day)	16.10		15.90		2.506	0.957	–	–	–
MICNR (g/g)	0.22		0.18		0.039	0.359	–	–	–
EMS (g/kg DOM)	49.20		47.70		0.934	0.883	–	–	–
SUN (mg/dL)	10.20		15.70		1.773	0.073	–	–	–
UUN (g/day)	16.30		26.40		1.159	<0.001	–	–	–
UEUN (g/day)	24.40		34.70		1.753	0.006	–	–	–
FNE (g/day)	21.20		21.80		2.164	0.837	–	–	–
NB (g/day)	30.50		32.50		2.448	0.594	–	–	–
EFNU (g/g)	0.40		0.36		0.034	0.336	–	–	–
NI (g/day)	83.60	117.00	88.40	121.00	–	–	–	0.001	0.912
NMIC (g/day)	38.10	28.50	32.20	31.60	3.109		0.803	0.167	0.202
MICNR (g/g)	0.46	0.24	0.38	0.27	0.048		0.595	0.022	0.255
EMS (g/kg DOM)	117.00	88.40	102.00	97.60	12.507		0.824	0.272	0.398
SUN (mg/dL)	12.00	16.90	9.77	16.60	1.253		0.360	0.010	0.496
UUN (g/day)	24.00	46.40	19.70	49.80	2.072		0.860	<0.001	0.158
UEUN (g/day)	37.30	61.40	32.40	58.30	2.164		0.153	<0.001	0.713
FNE (g/day)	29.50	35.00	34.80	36.80	2.953		0.315	0.293	0.597
NB (g/day)	16.90	20.20	21.20	25.40	3.566		0.253	0.415	0.871
EFNU (g/g)	0.19	0.17	0.23	0.21	0.034		0.265	0.507	0.972

^1^ NI = nitrogen intake; NMIC = synthesis of nitrogenous compounds in the rumen; MICNR = microbial nitrogen:nitrogen consumed ratio; EMS = efficiency of microbial synthesis; SUN = serum urea nitrogen concentration; UUN = urea urinary nitrogen; UEUN = urinary excretion of urea nitrogen; FNE = fecal nitrogen excretion; NB = nitrogen balance; EFNU = efficiency of nitrogen utilization. ^2^ LO = animals received 6 g/kg of BW of supplement with 150 g CP/kg DM; HI = animals received 6 g/kg of BW of supplement with 300 g CP/kg DM. ^3^ SEM = standard error of means. ^4^ IP = effect protein level in the supplement at initial pre-weaning; FP = effect protein level in the supplement at final pre-weaning; and IP × FP = interaction effect between supplementation levels at initial pre-weaning × final pre-weaning.

**Table 6 animals-15-02913-t006:** Metabolic profile of male Nellore calves under grazing system receiving different protein levels in the supplement at pre-weaning.

Item	Initial Pre-Weaning ^2^				SEM ^3^	IP	*p*-Value ^4^		
	LO		HI				Overall		
	Final Pre-Weaning						IP	FP	IP × FP
	LO	HI	LO	HI					
Glucose (mg/dL)	77.60		81.60		3.803	0.476	–	–	–
Triglycerides (mg/dL)	32.80		39.20		3.781	0.261	–	–	–
Total proteins (g/dL)	6.52		6.67		0.117	0.388	–	–	–
Albumin (mg/dL)	3.24		3.42		0.004	0.015	–	–	–
Globulins (mg/dL)	3.28		3.25		0.010	0.824	–	–	–
IGF-1 (ng/mL) ^1^	424.0		441.0		1.978	0.549	–	–	–
Glucose (mg/dL)	82.70	77.00	82.10	85.30	3.975		0.394	0.773	0.333
Triglycerides (mg/dL)	25.50	31.50	26.40	28.70	2.876		0.718	0.243	0.532
Total proteins (g/dL)	6.83	6.67	6.91	6.93	0.082		0.122	0.437	0.338
Albumin (mg/dL)	3.28	3.28	3.34	3.43	0.071		0.231	0.603	0.606
Globulins (mg/dL)	3.55	3.38	3.57	3.50	0.084		0.864	0.538	0.591
IGF-1 (ng/mL)	338.0	383.0	349.0	356.0	25.90		0.772	0.377	0.520

^1^ IGF-1 = insulin-like growth factor–1. ^2^ LO = animals received 6 g/kg of BW of supplement with 150 g CP/kg DM; HI = animals received 6 g/kg of BW of supplement with 300 g CP/kg DM. ^3^ SEM = standard error of means. ^4^ IP = effect protein level in the supplement at initial pre-weaning; FP = effect protein level in the supplement at final pre-weaning; and IP × FP = interaction effect between supplementation levels at initial pre-weaning × final pre-weaning.

**Table 7 animals-15-02913-t007:** Productive performance of male Nellore calves under grazing system receiving different protein levels at pre-weaning.

Item	Initial Pre-Weaning ^4^				SEM ^5^	IP	*p*-Value ^6^		
	LO		HI				Overall		
	Final Pre-Weaning						IP	FP	IP × FP
	LO	HI	LO	HI					
Final BW (kg) ^1^	191.0		193.0		1.41	0.471	–	–	–
Average daily gain (kg/day)	0.98		1.00		0.023	0.471	–	–	–
SFTL (mm) ^2^	1.38		1.54		0.124	0.659	–	–	–
SFTR (mm) ^3^	2.06		2.30		0.127	0.194	–	–	–
Final BW (kg)	249.0	250.0	256.0	253.0	5.29		0.409	0.912	0.697
Average daily gain (kg/day)	0.87	0.88	0.91	0.89	0.032		0.409	0.911	0.698
SFTL (mm)	1.54	1.66	1.62	1.90	0.123		0.265	0.183	0.578
SFTR (mm)	2.36	2.41	2.49	2.70	0.123		0.190	0.369	0.570

^1^ BW = body weight. ^2^ SFTL = subcutaneous fat thickness over the longissimus muscle. ^3^ SFTR = subcutaneous fat thickness over the rump. ^4^ LO = animals received 6 g/kg of BW of supplement with 150 g CP/kg DM; HI = animals received 6 g/kg of BW of supplement with 300 g CP/kg DM. ^5^ SEM = standard error of means. ^6^ IP = effect protein level in the supplement at initial pre-weaning; FP = effect protein level in the supplement at final pre-weaning; and IP × FP = interaction effect between supplementation levels at initial pre-weaning × final pre-weaning.

## Data Availability

The data consigned in this study are deposited in the official repository of the Federal University of Viçosa: http://www.locus.ufv.br/handle/123456789/20165 (accessed on 11 September 2025).

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
