# Peer review of "Do Protein Supplementation Levels Influence the Performance of Male Nellore Calves Under a Grazing System at Pre-Weaning?"

_animals, 2025, doi:10.3390/ani15192913_

Round 1
Reviewer 1 Report
Comments and Suggestions for Authors
This paper investigates the optimal level of supplemental feed for pre-weaned male Nellore calves under a grazing system. The abstract and conclusions are well-structured, and the background, objectives, materials, and methods are clearly described. However, the Results and Discussion sections would benefit from further refinement. The manuscript contains all the necessary scientific information and is logically written, making it suitable for publication after minor revisions.
Specific Suggestions:
Table 1: It is recommended to briefly describe the differences between the three types of forage listed.
Results Section: Consider including digestibility data directly in the main text.
Lines 352–355: Confirm whether this sentence refers to data presented in Tables 2 and 4.
Line 370: Confirm whether a statistically significant difference was observed in plasma urea nitrogen levels.
Lines 503–507: The authors appear to suggest that protein from the supplement is more digestible than milk protein. If this is the case, it would strengthen the argument to cite relevant supporting literature.
Author Response
Comments 1: Table 1: It is recommended to briefly describe the differences between the three types of forage listed.
Answer 1: The B and C forage corresponded to pasture consumed for animals during each digestibility assay, whereas the D forage corresponded to pasture consumed for animals throughout the experimental period.
Comments 2: Results Section: Consider including digestibility data directly in the main text.
Answer 2: Finally, the protein levels in the supplement did not affect the variables related to the total digestibility of the DM (0.694 and 0.514 g/g), OM (0.743 and 0.565 g/g), CP (0.740 and 0.664g/g), EE (0.874 and 0.711 g/g), apNDF (0.613 and 0.411 g/g), NFC (0.8 and 0.647 g/g) in Initial pre-weaning and Final pre-weaning respectively (Table 4).
Comments 3: Lines 352–355: Confirm whether this sentence refers to data presented in Tables 2 and 4.
Answer 3: Corrected. Sentence refers to data presented in Table 3.
Comments 4: Line 370: Confirm whether a statistically significant difference was observed in plasma urea nitrogen levels.
Answer 4: Yes. HI calves had higher (P < 0.05) nitrogen intake (NI), concentrations of SUN (serum urea nitrogen)
Comments 5: Lines 503–507: The authors appear to suggest that protein from the supplement is more digestible than milk protein. If this is the case, it would strengthen the argument to cite relevant supporting literature.
Answer 5: In addition, the best nutritive value and mass of forage in Initial pre-weaning allowed animals to be more selective to satisfy their potential nutrient utilization rates. In contrast, in Final pre-weaning, the higher CP digestibility in the HI treatment can be explained by the higher intake of CP via supplement, since supplements usually presented a higher digestibility than pastures (Moura et al., 2020).
Reviewer 2 Report
Comments and Suggestions for Authors
Dear authors,
Congratulations on this interesting mannuscript.
The manuscript is well written, with a clear structure and reader friendly. It is interesting, comprehensive and detailed. The methodology, analysis and presentation of results are scientifically sound and easy to follow. The results are presented in 6 tables and 2 figures, which are relevant and easy to understand. The reference list is recent and relevant. Self-citations are contained but not extensively.
In my opinion, the main body of the manuscript is strong, however, the introduction is weaker. Strengthening this section would help to further emphasize the importance of the study and better highlight the significance of the authors’ experiment and findings. Also, the discussion could benefit by identifying limitations of the study and suggesting directions for future research.
Simple summary: Simple, concise, written in non-technical language. Just one suggestion, at L20 the term “C4 grasses” should be changed or explained, since the simply summary section should be easily understood by the general audience.
Abstract: Concise, technical, summarizing the method used and the key findings.
Introduction: Short, well-structured and easy to follow. However, in my view this section appears “weak” compared with the rest of the manuscript. I have the following suggestions: 1) This section could be improved by including additional references, particularly L45-55, L60-65. 2) The authors could also expand on the health problems that protein imbalance can cause to calves, thus, highlighting even more the importance of their study. At L64-65 the authors mention the production cost and dietary inefficiencies but could also explain and emphasize potential risks such as impaired rumen development and kidney problems. 3) A strength of the experiment is also that it examines not only protein supplementation levels but also the correct supplementation period during pre-weaning. This should be highlighted more clearly in the objectives.
Materials and methods: Detailed, comprehensive, technically and scientifically sound. Could be reproduced by another researcher.
L126: it would be useful to explain why samples were collected every 13 days, to help the readers better understand this choice
Results: Again, detailed and comprehensive, supported by clear, detailed tables which are easy to understand and appropriate figures that illustrate the findings.
Discussion: Detailed, well written analysis and interpretation of the results. Comments on further improvement: 1) L540-3, how do the authors explain the contrast of their results compared to the other study, to what may this difference be attributed? 2) The discussion part does not address any gaps/ limitations of the study 3) Based on the interpretation of the results, it would be valuable to suggest directions for future research.
Conclusion: This section appears short given the extensive and comprehensive research and the multiple parameters studied. While the main finding is clearly stated, some key results could be also mentioned to support the significance of this extensive research.
Author Response
Comments 1: The manuscript is well written, with a clear structure and reader friendly. It is interesting, comprehensive and detailed. The methodology, analysis and presentation of results are scientifically sound and easy to follow. The results are presented in 6 tables and 2 figures, which are relevant and easy to understand. The reference list is recent and relevant. Self-citations are contained but not extensively.
In my opinion, the main body of the manuscript is strong, however, the introduction is weaker. Strengthening this section would help to further emphasize the importance of the study and better highlight the significance of the authors’ experiment and findings. Also, the discussion could benefit by identifying limitations of the study and suggesting directions for future research.
Answer 1: Authors: Thanks for the suggestions and recommendations to improve our manuscript. We worked hard to improve it.
Comments 2: Simple summary: Simple, concise, written in non-technical language. Just one suggestion, at L20 the term “C4 grasses” should be changed or explained, since the simply summary section should be easily understood by the general audience.
Answer 2: Authors: Thank you for the comment. The manuscript has been revised accordingly, and the changes can be found in linen 20. Replace “C4” by “Tropical”
Comments 3: Introduction: Short, well-structured and easy to follow. However, in my view this section appears “weak” compared with the rest of the manuscript. I have the following suggestions: 1) This section could be improved by including additional references, particularly L45-55, L60-65. 2) The authors could also expand on the health problems that protein imbalance can cause to calves, thus, highlighting even more the importance of their study. At L64-65 the authors mention the production cost and dietary inefficiencies but could also explain and emphasize potential risks such as impaired rumen development and kidney problems. 3) A strength of the experiment is also that it examines not only protein supplementation levels but also the correct supplementation period during pre-weaning. This should be highlighted more clearly in the objectives.
Answer 3: Authors: Thank you for the comment. The manuscript has been revised accordingly, and the changes can be found in lines 46 – 50; 61 – 67; and 80.
Comments 4: Materials and methods: Detailed, comprehensive, technically and scientifically sound. Could be reproduced by another researcher. L126: it would be useful to explain why samples were collected every 13 days, to help the readers better understand this choice
Answer 4: Authors: Thank you for the comment. It was added a required (lines 129-130).
Comments 5: Results: Again, detailed and comprehensive, supported by clear, detailed tables which are easy to understand and appropriate figures that illustrate the findings.
Answer 5: Authors: Thank you so much for the comment. We're working hard on this.
Comments 6: Discussion: Detailed, well written analysis and interpretation of the results. Comments on further improvement: 1) L540-3, how do the authors explain the contrast of their results compared to the other study, to what may this difference be attributed? 2) The discussion part does not address any gaps/ limitations of the study 3) Based on the interpretation of the results, it would be valuable to suggest directions for future research.
Answer 6: Authors: It was changed as suggested. However, If they included other references to better explain these results (lines: 539 - 546).
Comments 7: Conclusion: This section appears short given the extensive and comprehensive research and the multiple parameters studied. While the main finding is clearly stated, some key results could be also mentioned to support the significance of this extensive research.
Answer 7: Authors: Thank you so much for the comment. However, the results were not included to avoid repetition, as these values ​​are found in the results section. Additionally, this section is based on the theoretical variables, not the operational variables.
Reviewer 3 Report
Comments and Suggestions for Authors
The present study provides valuable insight into the effect of protein levels in the initial and final weaning period of Nellore claves by using a 2 x 2 factorial design. The hypothesis is sound and well. The language need minor editing to make some sentences clear.
- L 23- non detrimental performance, please rewrite the sentence for better clarity.
 - Is there control in the study without extra protein supplementation

- Keywords: have scope for improvement

- Effect of sequencing different -please rewrite for better clarity.
 - L92-96: Not clear sentence, please rewrite for better clarity. Please also mention clearly in one treatment, how many animals were studied, sample size ?
 - Table 2: please edit the values to their respective column

- Please mention the effect size in tables

Author Response
Comments 1: L 23- non detrimental performance, please rewrite the sentence for better clarity.
Answer 1: Corrected…Our results showed that decreasing in protein levels in the supplement from 300 to 150 g CP/kg DM does not detrimentally affect the performance and efficiency of grazing male Nellore calves.
Comments 2: Is there control in the study without extra protein supplementation
Answer 2: No negative control was used in the study.
Comments 3: Keywords: have scope for improvement
Answer: Improved. Biopsy; creep-feeding; IGF-1, metabolic profile; muscle fibers diameter; nitrogen utilization.
Comments 4: Effect of sequencing different -please rewrite for better clarity.
Answer 4: Improved. This study aimed to evaluate the effects of sequential supplementation with different crude protein levels on performance, nutritional and metabolic characteristics, and efficiency of nitroge-nous compounds use in grazing male Nellore calves at pre-weaning
Comments 5: L92-96: Not clear sentence, please rewrite for better clarity. Please also mention clearly in one treatment, how many animals were studied, sample size ?
Answer: Improved. Fifty Nellore suckling male calves non-castrated with an initial age of 3.5 ± 0.1 months and average initial body weight (BW) of 114 ± 2.4 kg, accompanied by their dams (average weight of 473 ± 8.4 kg and BCS of 4.5 on a scale of 1 to 9 points according to National Research Council (NRC) [15]), were randomly allocated into eight pastures of Urochloa decumbens (5.0 ha and 6.25 cow-calf/pasture), previously deferred.
The number of experiemental units was: two groups with 12 animals and two groups with 13 animals.
Comments 6: Table 2: please edit the values to their respective column
Please mention the effect size in tables
Answer: Adjusted
Reviewer 4 Report
Comments and Suggestions for Authors
Manuscript ID: animals-3859411
Type of manuscript: Article
Title: Do protein supplementation levels influence the performance of male
Nellore calves under a grazing system at pre-weaning? 
Authors: Marcos Rocha Manso, Luciana Navajas Rennó, Edenio Detmann, Mário
Fonseca Paulino, Sidnei Antônio Lopes, Nicole Stephane de Abreu Lima, Deilen
Sotelo Moreno *, Román Maza Ortega *
The authors evaluate the effects of sequencing different protein supple-mentation levels on performance, nutritional and metabolic characteristics, and efficiency of nitrogenous compounds use in grazing male Nellore calves at pre-weaning. They demonstrated that the decrease in protein levels in the supplement from 300 to 150 g CP/kg DM does not detrimentally affect performance, nutritional and metabolic characteristics in male beef calves under a grazing system at pre-weaning. Even though this work is interesting, several concerns were found in the whole manuscript and require the authors' major revision.
Please refer to the comments in the PDF file and revise accordingly before further consideration.

Author Response
Comments 1: Simplify the message; reduce repetition of “few studies” and clarify novelty compared with existing studies.
Answer 1: Improved. Many studies with beef calves in tropical conditions report increased performance in response to increasing the protein level in the supplement. However, the evaluating the use of supplements with an adequate protein proportion and ideal supplementation period for Nellore calves in low- or medium quality grasses can allow establishing supplementation strategies that optimize animal performance and improve bioeconomic efficiency.
Comments 2: Clarify why sequencing supplementation levels is biologically important; currently, the rationale is not well justified in the abstract.
Answer 2: Improved. EXPLICAR
Comments 3: Phrase “does not detrimental” should be corrected to “does not detrimentally affect.”
Answer 3: corrected
Comments 4: Add background on the economic importance of weaning weight in tropical beef production (e.g., market price, survival rates).
Answer 4: Improved
Comments 5: Include recent references (post-2020) on tropical forage limitations; the literature cited is dated.
-Too long sentence, please separate. Check other section too...
Answer 5: Improved
Comments 6: Rephrase redundancy – the manuscript repeats “protein is expensive” twice. Condense for clarity.
Answer 6: Adjusted.
Comments 7: Hypothesis is weakly stated. Specify whether you expect no difference or some difference in nitrogen metabolism but not growth.
-Too long sentence, please separate. Check other section too..
Answer: Improved. Therefore, we hypothesized that providing a lower level of protein supplementation does not detrimentally affect the productive and nutritional performance of beef calves in a tropical pasture, but rather the efficient use of nitrogen compounds
Comments 8: clarify whether randomization accounted for dam parity or milk yield, since maternal effects can bias calf performance.
Long sentence, need to be shorter.
Answer 8: Improved. Yes, they were considered and can be verified in the similar milk production of cows.
Comments 9: Indicate whether pastures differed in forage quality; blocking might be necessary.
Answer 9: All pastures presented similar forage quality, shade, and water quality and availability.
Comments 10: Include actual analyzed nutrient composition, not just ingredient proportions.
Answer 10: The analyzed nutrient composition is informed in Table 1.
Comments 11: Footnote formatting in Table 1 is confusing. Standardize with superscripts.
Answer 11: Adjusted.
Comments 12: Explain why chromium oxide was chosen instead of titanium dioxide, which is now more common and less toxic.
Answer 12: Chromium oxide was used because it was part of the research methods used. However, in recent studies, we have opted to use titanium dioxide.
Comments 13: spot urine sampling can be controversial. Justify why this was sufficient compared to total urine collection.
Answer 13: Spot urine sampling was sufficient compared to total urine collection, because the animals were in grazing conditions, where several total urine sampling limitations can arise, such as labour, animal stress, changes in animal behaviour, and packaging and storing collected urine.
Comments 14: Formula for Milk4% missing clear units. Clarify.
Answer 14: Corrected.
Comments 15: State whether calves were fasted before blood sampling; fasting affects glucose and metabolites.
Answer 15: Adjuted. Samples were collected without fasting.
Comments 16: Biopsy procedure raises ethical concerns; specify anesthesia/analgesia.
Answer 16: This procedure was carried out with the appropriate veterinary care, including the application of local anesthesia (Lidocaine 2%, LidoVet, Bravet, Rio de Janeiro, Brazil) and suturing after collection. The animals were monitored and treated with antibiotics and anti-inflammatory drugs, and the sutures were removed two weeks later.
Comments 17: Clarify why calves, not paddocks, were considered experimental units, given potential pseudoreplication from common pastures.
Answer 17: Calves were considered experimental units, because As the evaluations were focused on individual performance and these measure-ments were collected individually, as recommended by Detmann et al. (2016)...lines 105-107.
Comments 18: Consider reporting effect sizes (Cohen’s d) or confidence intervals, not just P-values.
Answer 18: Adjusted.
Comments 19: Milk lactose values in Table 2 look inconsistent (0.33 vs 0.99). Verify correctness.
Lactose values (0.33 vs 0.99) appear swapped; check.
Answer 19: corrected.
Comments 20: CP:DOM ratio units not defined (g/kg?). Clarify.
Answer 20: Adjusted
Comments 21: Emphasize biological interpretation: were calves already above CP requirements?
Answer 21: Adjusted.
Comments 22: SEM column missing decimal consistency.
Answer 22: Corrected
Comments 23: Figures 1 & 2: Legends unclear; add sample size per treatment.
Answer 23: Adjusted.
Comments 24: Strengthen comparison with NRC (2016) requirements; show whether diet CP was above/below requirement.
Answer 24: In this study, the average dietary CP content was 140.2 g of CP/kg DM for LO-LO and HI-LO calves and 187.1 g of CP/kg for LO-HI and HI-HI calves. These values differ from the CP content required by the ruminal microbiota (145 g CP/kg DM), to maxim-ize the consumption of fibrous carbohydrates (Detmann 2014), which may justify the similar intake of forage and apNDF digestibility between treatments.
Comments 25: Expand on why higher CP intake did not translate into higher ADG. Possible energy–protein imbalance?
Answer 25: Yes. This lack of difference may be attributed to possible energy–protein ratio imbalance.
Comments 26: Discuss environmental implications of excess N excretion (urinary urea → ammonia pollution).
Answer 26: which can lead to an increased the risk of nitrate (NO3−) leaching and ammonia (NH3) and nitrous oxide (N2O) emissions (FAO, 2002).
Comments 27: Correct grammar and provide mechanistic explanation: microbial protein supply vs rumen degradable protein balance.
Answer 27: Decreasing the CP level in the supplement did not have a detrimental effect on the nitrogen balance and EFU in the animals, which may suggest that the PB flow of 150 g/kg DM via supplement was sufficient to allow an adequate balance of degradable protein in the rumen, favoring the contribution of microbial protein.
Comments 28: Acknowledge that study duration (156 days) might not be long enough to detect small differences in growth.
Answer 28: adjusted
Comments 29: Consider citing fetal programming studies to contrast prenatal vs postnatal muscle fiber development.
Answer 29: Adjusted.
Comments 30: Avoid overgeneralization; results apply only to Nellore calves under creep feeding at 6 g/kg BW.
Answer 30: The decrease in protein levels in the supplement from 300 to 150 g CP/kg DM to Nellore calves under creep feeding at 6 g/kg BW does not detrimentally affect the performance, nutritional and metabolic characteristics, or efficiency of using nitrogenous compounds in male beef calves at pre-weaning under a grazing system. Therefore, supplementation with 150 g CP/kg DM in the amount of 6 g/kg BW for male beef calves at pre-weaning is recommended.
Comments 31: Suggest caution – economics and environmental sustainability should also guide recommendations.
Answer 31: Adjusted. The authors consider it pertinent to draw conclusions based on the hypothesis. Thus, economics and environmental sustainability were not the subject of study and may be considered in future work.
Comments 32: Update citations beyond 2020; currently most are older.
Answer 32: Adjusted.
Round 2
Reviewer 4 Report
Comments and Suggestions for Authors
The authors have provided a clear response, and no further suggestions are required.